# Evolutionary Qβ Phage Displayed Nanotag Library and Peptides for Biosensing

**DOI:** 10.3390/v15071414

**Published:** 2023-06-22

**Authors:** Augustin Ntemafack, Aristide Dzelamonyuy, Godwin Nchinda, Alain Bopda Waffo

**Affiliations:** 1Department of Biochemistry and Molecular Biology, Indiana University School of Medicine, 635 Barnhill Dr., Indianapolis, IN 46202, USA; antemafa@iu.edu (A.N.); ardzela@iu.edu (A.D.); 2Laboratory of Vaccinology and Biobanking, CIRCB BP 3077 Messa, Yaoundé P.O. Box 3077, Cameroon; nsehleseh@gmail.com; 3Department of Pharmaceutical Microbiology and Biotechnology, Nnamdi Azikiwe University, Awka 420110, Nigeria; 4African Center of Excellence for Clinical and Translational Sciences (ACECTS), Yaoundé P.O. Box 13591, Cameroon

**Keywords:** biosensor, biotin-tag, transducer

## Abstract

We selected a novel biotin-binding peptide for sensing biotin, biotinylated proteins, and nucleotides. From a 15-mer library displayed on the RNA coliphage Qβ, a 15-amino acid long peptide (HGHGWQIPVWPWGQG) hereby referred to as a nanotag was identified to selectively bind biotin. The target selection was achieved through panning with elution by infection. The selected peptide was tested as a transducer for an immunogenic epitope of the foot-and-mouth disease virus (FMDV) on Qβ phage platform separated by a linker. The biotin-tag showed no significant influence on the affinity of the epitope to its cognate antibody (SD6). The nanotag-bound biotin selectively fused either to the C- or N-terminus of the epitope. The epitope would not bind or recognize SD6 while positioned at the N-terminus of the nanotag. Additionally, the biotin competed linearly with the SD6 antibody in a competitive ELISA. Competition assays using the selected recombinant phage itself as a probe or transducer enable the operationalization of this technology as a biosensor toolkit to sense and quantify SD6 analyte. Herein, the published Strep II nanotag (DVEWLDERVPLVET) was used as a control and has similar functionalities to our proposed novel biotin-tag thereby providing a new platform for developing devices for diagnostic purposes.

## 1. Introduction

Antibodies and serological testing are the gold standard for detection and sensing specific targets [1,2,3,4,5]. Detection, quantification, or sensing of specific antibodies is a major component of several point of care kits currently used in clinical diagnosis [6,7]. Using experience gained from recently mapped epitopes through our novel evolutionary RNA phage display system, we are now extending this technology to targeting ligands, materials recognizing peptides, or biological structures [8]. Evolutionary RNA phage display is a powerful emergent nanotechnology for screening [9,10,11,12,13], evolving, and sensing peptide libraries [8] against a broad range of desired specific targets including biotin, gold, and other useful nanoparticles. Affinity targets can promote detection, sensing, and purification of recombinant proteins from complex mixtures [14,15,16,17,18]. Additionally, any target peptide can be a complement for antibody sensing and quantification while fused with a specific epitope through a linker on the recombinant RNA phage display platform. On the recombinant RNA phage, a 15-mer peptide target does not alter the charge or the structure but competes with the binding size due to its proximity. Due to their smaller size, they do not abrogate the biological function of the engineered epitope bearing the inserted peptide. The platform herein is the RNA coliphage Qβ which is a positive-sense single-stranded RNA bacteria-infecting phage, belonging to the family of *Fiersviridae* [19].

Qβ phages are found throughout the world in bacteria associated with sewage and animal feces [20,21,22,23]. Each infectious Qβ phage is about 25 nm in diameter [24]. Qβ is made up of four genes within a 4220 nucleotides genome which encodes for a subunit II (β) replicase, a major coat protein (Cp), a maturation protein (A2 or MA2), and a minor coat or read-through protein (MCPA1 or A1) [25,26,27,28,29,30]. The life cycle of the Qβ phage has been previously elucidated and starts with the adsorption of the phage on the bacteria’s F+ via the A2 protein, followed by the injection of the RNA into the cytosol [31,32]. The A1 protein shares the same initiation codon with the Cp and is produced during translation, when the Cp stop codon UGA triplet is suppressed by a low level of ribosomal read-through and incorporation of tryptophan at the coat protein termination signal [33,34].

The coliphage Qβ marks a key point in evolutionary technology with its RNA-dependent RNA polymerase (RdRp). Unlike any other polymerase, RdRp has high mutation rates in the order of 10^−4^ which is strategically very important in the generation of a viable, adapted, and evolved RNA Qβ hybrid or recombinant phage. We discovered several desirable features that make this RNA phage suitable for recombinant surface engineering for probe display. These include the fact that (i) the A1 minor coat protein can be extended by 50 amino acids without affecting its function, (ii) the A2 protein can initiate the Qβ hybrid phage infection while binding to an immobilized target, and (iii) the replicase promiscuity is a key to any quick adaptability, thereby providing a unique opportunity in evolutionary technology through a novel biopanning strategy optimized by our group [8]. In this report, we have utilized plasmids with the full cDNA of Qβ phage for the construction of expression cassettes as vectors not only for Qβ production but equally for recombinant phages after surface engineering of a peptide library and/or a FMDV epitope (5 amino acids) with and without biotin peptide (15 amino acids). Subsequently, the roles of the surface engineered recombinant phages are assessed in sensing, evolution, concentration, detection, and quantifying biotin fuse with a linker (seven amino acids). Altogether, 27 amino acids were fused to the recombinant phage genome and exposed on the surface of its capsid in an accessible and assessable way by their cognate antibodies.

Our aim was to select a novel biotin-binding peptide from a recombinant phage library after fusion through a linker coupled to an FMDV epitope to provide a reconstructed peptide on a recombinant Qβ platform that would quantify anti-FMDV antibodies. A library of 10^9^ recombinant A1 fusion proteins was created encoding 15 contiguous amino acids at random. Utilizing our novel panning strategy involving elution by infection instead of acidic, a peptide was identified selectively binding only biotin, biotinylated oligonucleotide, or protein. The washed non-binding peptides were tested against gold and other nanoparticles. The candidate peptide with the best specific binding ability was engineered via a linker with the FMDV epitope for a competitive ELISA analysis. This produced several different promising recombinant phage variants selectively binding to biotin in competition with anti-FMDV antibodies. We next optimized and tested the site of insertion of the target peptide and its function in affinity tagging of biotin successfully. Thus, the biotin recognizing peptide tag could be used to detect, sense, and quantify biotin or biotinylated entities (proteins, RNA, or DNA).

## 2. Materials and Methods

### 2.1. Materials

#### 2.1.1. Reagents, Facilities, and Providers

Media used for bacteria culture were purchased from Fisher Scientific (Pittsburgh, PA, USA). Restriction enzymes, T4 DNA ligase, Taq polymerase, and alkaline phosphatase calf intestinal (CIP) were purchased from New England Biolabs (Ipswich, MA, USA). The enzymes for reverse transcription, RNA preservation, and RNase free water were purchased from Promega (Madison, WI, USA). The phage dialysis cassettes were purchased from Thermo-Fisher Scientific (Rockford, IL, USA). The viral RNA extraction, DNA gel extraction, and plasmid mini, maxi, and midi preparations were conducted with Qiagen kits (Valencia, CA, USA). Antibodies were purchased from Sino Biological (Wayne, PA, USA). Genes and oligonucleotides were synthesized by Eurofins (Louisville, KY, USA). Reagents for ELISA, Dot blotting, and the 1x Roti block were purchased from Carl ROTH (Karlsruhe, Germany).

Chemicals and other reagents (i.e., RbCl and CaCl2) were purchased from Sigma-Aldrich (St Louis, MO, USA). All DNA sequencing reactions were performed by AZENTA (South Plainfield, NJ, USA). Our standard ELISA testing was conducted with ARVYS (Trumbull, CT, USA). The cryo-EM imaging was performed at the Indiana University School of Medicine Core Facility (Indianapolis, IN, USA) that also provided reagents. The phage library preparation was done at the Chemical Genomic Core Facility (CGCF) of Indiana University School of Medicine (IUSM).

#### 2.1.2. Microorganisms

The bacteria *E. coli* DH5α and MC1016 from Invitrogen (Grand Island, NY, USA) were used to propagate and maintain plasmids for subcloning and recombinant plasmids. *E. coli* HB101 and DH5α were used to grow and maintain pBRT7Qβ or pQβ7 plasmids and all their recombinant derivatives. Three different indicator *E. coli* bacteria from ATCC (Manassas, VA, USA) were used for phage production and titration: K12, Hfrh, and Q13. The *E. coli* bacteriophage Q-βATCC 23631-B1 from ATCC (Manassas, VA, USA) was used as a positive control during experiments.

#### 2.1.3. Plasmids and Oligonucleotides

Plasmids pBRT7Qβ [35], pQβ7, and pQβ8 [36] were obtained from Professor Weber and Professor Kaesberg groups, respectively, and used to construct variants to ease genetic modifications in our group [8,9,10,11,12]. These plasmids, pBRT7Qβ having 7489 bp (from 1 to 7489 when restricted with SmaI) and pQβ8 having 7393 bp (from 1 to 7393 when restricted with SmaI endonuclease), were used for this work because they both contain the entire cDNA of phage Qβ with different orientations. All nanotag genes with A1 were constructed by fusion PCR with synthetic oligonucleotides as presented in Table 1.

#### 2.1.4. Protein A1 and Nanotag Peptide Fusion 3D Computer Simulation

The fusion sequences of the minor coat protein A1 and engineered with the nanotag proteins were modeled with the Raptor X web server (doi: 10.1038/nprot.2012.085) using template-based modeling. The models were then transformed to view the protein backbone and highlight the secondary structures using MolGro molecular viewer. The structures were next aligned to understand the differences among the protein models.

#### 2.1.5. Analysis of Protein A1 and Nanotag RNA Secondary Structures

Amino acid sequences for the codon-optimized nanotag, FMDV epitope, and chimeric nanotag-FMDV were designed and analyzed for an appropriate cloning in expression cassettes at the phages’ A1 C-terminus using the sequence analysis software DNA strider while taking into consideration the secondary structure of the RNA of the recombinant coliphage as previously reported [18,23]. The RNA secondary structure of the designed A1-nanotag-epitope regions including a (GGSGGS)2 linker was analyzed using the RNA-Fold software.

### 2.2. Methodology

#### 2.2.1. Construction of the RNA Phage Vector Library

A combinatory library made of three oligonucleotides was conducted using a protocol previously described [8]. The 15-mer synthetic library was obtained by combining three oligomers ABW1, 2, and 3 as presented elsewhere using the method of heat and cool annealing overnight in low salt buffer [8]. The library contained the randomized middle part, flanked at both ends of the complementary sequences, representing the linker sequence within the plasmids pQβ7/pQβΔA1, which were previously constructed [8]. For the construction of a vector for hybrid phage expression, a total volume of 20 µL with a maximum of 10 ng of synthesized oligomers was all phosphorylated, annealed, and ligated into 10 ng of linearized pQβ8. We have recently optimized the protocol of the oligonucleotide phosphorylation with ^32^P and gel analysis. Prior to this ligation, pQβ7 was restricted with AflII and NsiI, dephosphorylated, and gel-purified.

#### 2.2.2. Expression of the RNA Phage Vector Display Library

A volume of 20 µL of each ligation mixture was used to transform either *E. coli* HB101 or DH5α laboratory-prepared competent cells. The competent cells were prepared from *E. coli* HB101 using the RCl method as described elsewhere [8] with a slight modification. Briefly, a single colony from a fresh overnight 2YT-agar plate was inoculated into 50 mL of 2YT broth (16 g bacto tryptone, 8 g bacto yeast extract, and 5 g NaCl all in 1 L) and incubated at 37 °C while shaking overnight at 220 rpm. Approximately, 1/100 of the culture was inoculated into 50 mL of fresh 2YT broth to allow growth to the early log phase (OD600 reading of 0.5–0.7) and kept on ice for 30 min followed by centrifugation at 4000 rpm at 4 °C. All subsequent preparation steps were carried out on ice. The resuspended pellet was gently washed with 7 mL of solution 1 (0.1 M MOPS and 0.1 M RbCl, pH 7) and centrifuged at 4000 rpm for 10 min at 4 °C. The supernatant was discarded, and the pellet was resuspended in 7 mL of solution 2, (0.1 M MOPS, 0.1 M RbCl, and 0.5 M CaCl2, pH 6.5). The suspension was kept for 30 min followed by centrifugation in the conditions mentioned above. The supernatant was discarded, and the pellet resuspended in 2 mL of solution 2. The competent cells were aliquoted at 200 µL each in vials and used. For transformation, a volume of up to 20 µL of the ligated library was added to 200 µL of competent cells and kept on ice for 15 min. The mixture was heat-shocked at 42 °C for 45 s and kept on ice for 2 min. A volume of 3 mL of 2YT media was added to the mixture and incubated at 37 °C in a shaking incubator for 45 min at 220 rpm. The entire culture was spread on 2YT agar supplemented with ampicillin at a final concentration of 400 µg/mL and incubated at 37 °C ±overnight (100 µL per plate for a total of 30 plates per single transformation). Individually, all colonies were picked and inoculated into 3 mL of 2YT supplemented with ampicillin (100 µg/mL) to promote single growth. The cultures were incubated at 37 °C for 8 h, pooled together to form (a library) 1 l, precipitated with PEG (80 g/L) and NaCl (29.5 g/L) at 4 °C overnight, and spun at 4500 rpm for 30 min. The supernatant was discarded, and the pellet containing the phage was resuspended in 300 mL of phage buffer (10 mM Tris HCl pH 7.5, 1 mM MgCl2, 100 mM NaCl, and 10 mg/L gelatin with 1/5 of the volume of phage suspension after amplification). The resuspended pellet was centrifuged at 10,000 rpm for 30 min at 4 °C. The supernatant containing the phage was recovered, and the pellet was discarded. A volume of 20 mL of phage was obtained from a 1 l pooled culture. At this final stage, all the phages in a volume of 20 mL each were re-pooled and precipitated. A final volume of 5 mL of phage buffer was used to resuspend the pooled precipitated transformants that were obtained and dialyzed against the phage buffer representing the library of phages.

#### 2.2.3. Construction of a Simple RNA Phage Display Vector for Tag Peptide Probe

An expression cassette of the designed RNA phage display system was used to design primer sequences with the forward primer flanked with Bpu10I (portion of the phage cDNA between 1711 and 1760 bases) and the reverse as per Table 1 with NsiI. Additionally, the reverse primer was flanked by a unique restriction enzyme sequence for clone analysis prior to sequencing. A special PCR program was performed with the pQβ7 plasmid as the DNA template. The PCR products were run in 1.2% low-melting agarose gel electrophoresis, and the DNA fragment bands were isolated and purified using the Qiagen kit. For the cloning procedure into pQβ7 plasmid, Bpu10I and NsiI restriction sites were used. The PCR fragments (inserts) were then separately digested using the same restriction enzymes as the vector mentioned above without dephosphorylation. The digested fragments were run in a 1.2% low-melting agarose gel electrophoresis, and bands of ~640 bp were isolated and purified. The digested vector was then dephosphorylated with CIP for an additional 1 h at 37 °C. The digested dephosphorylated vector was run in a 1% low-melting agarose gel electrophoresis, and the DNA band of ~7000 bp was extracted and purified using the Qiagen kit. Each restricted PCR fragments was ligated into the restricted and dephosphorylated pQβ7 plasmid at 16 °C overnight with T4 DNA ligase (30 ng of plasmid with 12 ng of insert in 20 µL volume). The total ligation mixture was used to transform *E. coli* MC1016 following the previous protocol with modification. The 3 mL of transformants was spun down, at 4500 rpm for 10 min, suspended with 300 µL of 2YT, plated on 2YT-agar supplemented with ampicillin (100 µg/mL), and incubated overnight. For screening, 10 clones from each construction were used to prepare DNA using the Qiagen kit. The DNA obtained from each clone was analyzed by restriction digestion using the unique enzyme from the design. The digested recombinant DNA was run on a 1.2% agarose gel electrophoresis to confirm the validity of the clone. The valid clones were sequenced to confirm the presence of the insert within the frame at the end of the A1 minor coat protein. The positive clones were used for retransformation of the *E. coli* HB101 competent cells to produce phages and amplify plasmids.

#### 2.2.4. Bacteria Transformation for Recombinant Phage Expression

For transformation, 5 µL of the sequenced recombinant pQβ8 plasmid was added to 200 µL of competent *E. coli* HB 101 cells and kept on ice for 15 min. The mixture was heat-shocked at 42 °C for 45 s and kept on ice for 2 min. A volume of 3 mL of 2YT media was added to the mixture and incubated at 37 °C in a shaking incubator for 45 min at 200 rpm. The culture was spread on 2YT agar supplemented with ampicillin (100 µg/mL) and incubated overnight at 37 °C. Into 3 mL of 2YT supplemented with ampicillin (200 µg/mL), 2 colonies from each transformation were inoculated. The culture was incubated at 37 °C for 5 h, transferred into 1 L of 2YT supplemented with ampicillin (400 µg/mL), and incubated at 37 °C overnight at 200 rpm. The resulting culture was precipitated with PEG (80 g/L) and NaCl (29.5 g/L) at 4 °C overnight and spun at 4500 rpm for 30 min. The supernatant was discarded, and the pellet was resuspended in 200 mL phage buffer. The resuspended pellet was centrifuged at 10,000 rpm at 4 °C for 30 min. The supernatant containing the phage was recovered, and the pellet was discarded. The phage titer was determined by serial dilution and an agar overlay assay.

#### 2.2.5. Checking for Recombinant Plasmid Phage Production

The supernatant of the recombinant plasmid pQβ8 in E. coli HB 101 grown overnight was analyzed for phage expression using the agar overlay for spot test. This test was conducted as previously described [8]. An *E. coli* Q13 or Hfrh bacteria culture was grown to the log phase (OD600 of 0.5–0.7). A volume of 100 µL was added to 3 mL of YT-Top-agar, and the mixture was poured on the surface of 1.5% nutrient agar plates. The plates were left to solidify at 37 °C for a few minutes. Thereafter, 10 µL of the phage suspension was dropped on the solidified top-agar plates. The plates were incubated at 37 °C for 8–12 h and examined for a possible lysis of the *E. coli* Hfrh lawn where the droplet of phage suspension was placed.

#### 2.2.6. Phage Scaling and Titration

##### Scaling of Phages from *E. coli* HB101

A volume of 3 mL of *E. coli* Hfrh overnight culture was inoculated in 300 mL of TGY medium (15 g bacto tryptone, 1 g bacto yeast extract, 8 g NaCl, 1 g glucose, and 0.2 g CaCl_2_ all in 1 l). The culture was incubated at 37 °C for 3 h while shaking at 200 rpm and infected with 200 mL of the phage suspension obtained from *E. coli* HB101. It was then allowed to stand for 30 min at 37 °C (without shaking) prior to incubation for 5 h while shaking at 160 rpm. The 500 mL phage culture was used to infect 1500 mL of the host cell (*E. coli* Hfrh) preincubated at 37 °C for 3 h at 200 rpm. After infection, the culture was incubated under the same conditions mentioned above. Thereafter, the phage culture was kept at 4 °C and precipitated overnight by adding PEG8000 (80 g/L) and NaCl (29.5 g/L) followed by centrifugation at 4 °C for 30 min at 4500 rpm. The supernatant was discarded, and the pellet was resuspended in 400 mL of phage buffer. The suspension was centrifuged at 10,000 rpm. The supernatant containing the phages was recovered, and the pellet was discarded. The phage was precipitated with PEG/NaCl at 4 °C overnight and pelleted using the above conditions. The supernatant was discarded, and the resulting pellet was resuspended in 40 mL of phage buffer (10 mM Tris HCl pH 7.5, 1 mM MgCl2, 100 mM NaCl, and 10 mg/L gelatine), and the suspension was centrifuged at 12,000 rpm at 4 °C for 30 min. The supernatant was recovered and dialyzed at 4 °C overnight in a dialysis cassette against phage buffer (without gelatin). The phage suspension was stored at 4 °C and the titer determined.

##### Phage Titration

The titer of phages was determined using agar overlay assay as previously described [8] with a slight modification. Briefly, solid agar (1.5% agar) was autoclaved, poured into a Petri dish, and allowed to solidify at room temperature. The phage solution was subjected to a serial dilution in TGY broth with a 1:1000 factor, and 100 µL of each dilution was added separately to a test tube containing 100 µL of *E. coli* Q13 at log phase. Subsequently, 3 mL of TGY soft agar (TGY with 0.5% agar) was added to the mixture, mixed well, and poured onto the solid agar contained in a Petri plate. The culture medium was allowed to solidify at room temperature and incubated at 37 °C for 6 h. Thereafter, the plates were checked for a possible formation of clear zone (plaques). The plate with countable plaques was examined, and the corresponding dilution was used to determine the phage titer expressed in plaque forming unit per ml (pfu/mL).

##### Panning Procedure

The recombinant biotin, biotinylated RNA, or peptide in coating buffer (0.1M NaHCO_3_ pH 9.6) was coated in microtiter plates at 4 °C overnight. After blocking the plate with 3% BSA in PBS, a volume of 200 µL with a titer of 10^9^ pfu/mL phages diluted from the stock was added into each well, and the biopanning protocol reported elsewhere proceeded [8].

#### 2.2.7. Phage Characterization

##### Molecular Identification

Phage RNA was extracted using the Qiagen kit. The isolated RNA was checked on 0.8% agarose gel electrophoresis and quantified using a nanodrop spectrophotometer. A total volume of 10 µL of reaction mixture (200 ng RNA, 1 µL reverse primer, and nuclease-free water) was subjected to 7 min of primer annealing followed by the addition of 15 µL of the mixture (buffer: 5×, 5 µL, dNTPs: 2.5 µL, reverse transcriptase 1 µL, RNase inhibitor: 1 µL, and nuclease-free water 5.5 µL). The reaction was subjected to reverse transcription in a thermocycler. A volume of 5 µL of the obtained cDNA was used as cDNA template in a polymerase chain reaction (PCR) for amplification. A total of 50 µL reaction mixture containing HF-buffer (5×, 10 µL), nuclease-free water (31.5 µL), forward primer (1 µL), reverse primer (1 µL), dNTPs (1 µL), and polymerase (0.5 µL) was subjected to 25 cycles of amplification in a thermocycler. The size of the PCR product was analyzed by agarose electrophoresis. The appropriate cDNA fragment was electrophoresed in 1.2% low melting agarose gel. Thereafter, the fragment was extracted and purified using the Qiagen gel extraction kit and subsequently sequenced.

##### Dot Blotting

For analysis by dot blotting, 10–20 µL of phages with a titer of 10^12^ pfu/mL was spotted on the nitrocellulose membrane, allowed to dry for 30–45 min, and blocked with 1x roti block. The blocked membrane was then probed with the appropriately diluted antibody or protein (anti-His-tag 1:500, anti-His-tag-HRP 1:1000, streptavidin-HRP 1:1000, and biotin-HRP 1:2000; HRP is the horseradish peroxidase). The specific recognition of the tag probe was revealed with HRP-conjugated rabbit anti-mouse IgG diluted 1:1000 in 1× roti block. The bound conjugate was detected using 1-step ultra TMB blotting solution, and the HRP reaction was stopped by washing with molecular grade pure water.

##### ELISA

The appropriate recombinant phages were diluted separately in PBS so that 100 µL containing 10^7^ phage particles was added to high-binding 96-well flat-bottom microsorb ELISA plates and incubated at 4 °C overnight. The following day, plates were washed 3x with PBST (PBS with 0.05% Tween-20) and blocked either with 3% BSA or 1× Roti block for 1 h at 37 °C. The coated plates were then probed with graded doses of monoclonal antibodies diluted in 2% BSA at concentrations of 1000, 100, 10, 1, and 0.1 ng/mL. A volume of 100 µL of the diluted antibodies was added per well and incubated for 2 h at 37 °C. Unbound antibodies were removed by washing 5× (198 µL/well) with PBST. The reactivity of the monoclonal antibodies with the recombinant phages was probed HRP-conjugated Goat anti-mouse IgG diluted at 1:4000 in 0.1× Roti block. The bound conjugate was detected using ABTS substrate, and the HRP reaction was stopped by adding 100 µL of stop solution according to the manufacturer’s protocol. The colorimetric signal was measured at 405 nm using a multiscan FC microplate reader.

##### Microscopy

Scanning electron (EM) and cryo-electron microscopy (Cryo-EM) were used to confirm the morphology of the recombinant phages. EM was performed as previously described elsewhere [8]. Briefly, 5 µL of pure phages at a concentration of 10^15^ pfu/mL was loaded onto a carbon-carbon coated formvar grid and kept for a few minutes. A few drops of aqueous uranyl acetate were added to the preparation, and the slide was observed under the electron microscope (JEOL 1200EX, Tokyo, Japan). Cryo-EM was carried out following the method developed by Liu [37] with slight modification. Briefly, 3 µL of pure phages was loaded onto a glow-discharged grid and blotted using Vitrobot Mark IV. The grid was frozen in liquid nitrogen and transferred to a transmission electron microscope. The sample was analyzed at 300 kV, and the image was taken at a magnification of 59,000×.

## 3. Results

### 3.1. RNA Qβ Phage Displaying 15-Mer Library

An RNA phage (Qβ) was chosen for developing an ORF that permits the display of a fused insert at the C-terminal of the A1 because this phage has a robust growth rate and its replicase (RdRp) promiscuity favors directed evolution [34]. This is in addition to its functional flexibility permitting numerous A1 truncations. The design and principle of phage display library insertion on Qβ cDNA are depicted in Figure 1. The first challenge was to obtain the library of RNA phages with at least 10^9^ pfu/mL variants. Additionally, pooling 10^9^ pfu/mL of phages from small ligation (10 ng of insertion) with a single round of amplification with *E. coli* K12 gave the appropriate titer for use in biopanning. The library was successfully fused at the end of the A1 minor coat protein gene sequence terminating in two natural opal and ochre stop codons TGA and TAA, respectively. The restriction enzyme sequences flanking the library (ABW1, Figure 1) were used to insert the randomized sequence at the end of the A1 gene. Using the restriction enzyme sites Afl II (position 2159) and Nsi I (position 2350), 192 nucleotides were deleted at the end of A1. The deletion of A1 and extension with the library were exploited to produce variants for this evolutionary library. Additionally, a specific Shine Dalgarno (SD) sequence (TAAGGAGG) was added to the intact intercistronic region (position 2339) after the insertion cassette thereby improving the phage titer. These results demonstrated that the truncated A1 can accommodate the library and confirmed the key role of the intercistronic region between the A1 and the replicase genes in recombinant Qβ phage production [10]. Two plasmids were constructed with the modification both containing the cDNA of Qβ namely the pQβAd2 [8], pQβAd2SD with A1 truncated, and an added SD, respectively. The plasmid pQβAd2SD produced recombinant phages with a titer close to the wt, while the pQβAd2s’ expression titer was three-fold lower. A recombinant phage library with a titer up to 10^9^ pfu/mL of phages was successfully obtained which was enough for subsequent selection through our optimized panning strategy [8].

### 3.2. Biotin and Biotinylated RNA/Peptide Binding Sequences

The goal of this study was to develop biotin-binding peptides using an RNA phage display system and test whether the same peptides could be used as probes to detect and quantify biotin or biotinylated entities. Captured biotin and biotinylated entities were separately immobilized on a plate, and the ORF phage library was enriched by binding to the reaction platform. The recombinant phage particle with the A1 protein extended by the library can be anchored through the interaction between the specific probe displayed and the biotin bound target. When anchored on its target after several washes, the recombinant phage is amplified through the A2 by adding a fresh log phase (OD600 = 0.7) *E. coli* K12 culture [8]. The recombinant phages were then eluted by infection and used to bind another biotin target. Six rounds were performed, and phages obtained after each round were sequenced. The predominant sequence binding biotin or biotinylated entities was HGHGWQIPVWPWGQG with the IPVW motif present in weaker binders. We reasoned that the IPVW motif gained fitness and was selected, amplified, and enriched to the final peptide binding biotin.

### 3.3. Biosensor with Biotin-Binding Peptide

#### 3.3.1. Design and Generation of Recombinant Plasmids

To design and generate plasmid vector variants of the probe, linker, or transducer, pQβ7 was used. We previously showed that up to 100 amino acids could be inserted, fused, and exposed on the A1 platform of Qβ recombinant phage without affecting phage viability and propagation. Additionally, we have reported the production of recombinant phages with plasmids containing a truncated A1 with a fusion of heteromeric peptides separated by the linkers. Similarly, several plasmids were constructed with the design as presented in Figure 2 showing pQβStrep, pQβBiot, pQβBiotFMDV, pQβBiotFMDV6His, pQβStrepFMDV with the streptavidin, biotin tags, the biotin tag with FMDV epitope, the biotin tag with FMDV epitopes fused with His-tag, and the streptavidin tag with FMDV epitope, respectively. This design was conducted with amino acid sequences positioned as shown in our illustration in the expression cassette (Figure 2) which were fused in frame to A1 to generate recombinant phages. Separately, the peptide tag, linker, epitope, and His-tag were all obtained by PCR within the reverse primer (Table 1) while the forward primer was part of the phage cDNA sequence. Previously, we conducted a comparative three-dimensional modeling of the A1 protein fused with our displayed peptides.

#### 3.3.2. Modeling of A1 with Peptide Inserted at the C-Terminus

The 3D structures analyzed were for the A1 in Qβ wt, QβStrep, QβBiot, and QβAu, respectively. Any change in the A1 structure and conformation while harboring a fusion peptide is shown by the green arrow (Figure 3). The N-terminus of the minor coat protein displaying peptides was not significantly changed in comparison to the wt. Structurally, the A1-bearing additional peptide conserved its α-helixes and β-sheets in all models (Figure 3). Looking at the C-terminal part of A1 with the insert, only minor rotations were observed. All additional peptides to A1 were confirmed to be exposed around the β-sheets and pointing to the outer surface of the capsid as shown in Figure 3B–D. This result was similar to previous 3D models of A1 fused proteins reported earlier in our group [9,10,11,12,13] which have been developed to efficiently engineer phage platforms with surfaces accessible to various entities (peptide and chemical).

#### 3.3.3. Secondary Structure of 5′ Untranslated RNA Region of the Replicase

The software program RNA-Fold was used to predict the secondary structure of the RNA 3′ untranslated region of the replicase after any tag and the Shine Dalgarno sequence insertions. The region between the A1 stop codons (2331) and the replicase start codon (2353) was checked for the availability of the start codon upon insertion of 100 to 200 nucleotides and different constrains of the newly formed hairpin with the stable tetraloop motif on this region. Any 5′ replicase domain constrain too close was optimized to fit the known secondary structure model for the two distal domains of the Qβ RNA. The plasmid pQβBiotFMDVHis was found to contain those constrains and produced a low titer of phage 10^4^ pfu/mL. The titer was brought to 10^7^ pfu/mL after a C (CAC) substitution to U (CAU) at the third position. The RNA secondary structure prediction has therefore contributed to the optimization of the display system on this single-stranded RNA phage.

#### 3.3.4. Recombinant Phage Vector Construction Strategy and Genetic Analysis

The reverse primer containing each gene fusion of the designated tag, the partial A1, the linkers, and the natural stop codons of the phage at this region were used with the common forward primer to generate each fragment shown in Figure 4 using pQβ7 as a template. The results suggested that gene fusion was obtained for each tag with two different reverse primers. These fragments are critically important in the construction of the recombinant plasmids for biosensor expression using the template vector. The small fragment products of PCR contained the truncated A1 and were previously tested to produce viable phages [8]. The large fragments were cloned between Bpu 10I and Nsi I, while their small counterparts were between Afl II and Nsi I, respectively. Two different genetic constructions were made with each tag and successfully analyzed with a restriction enzyme. The recombinant plasmid bearing the tag sequence was recognized within the A1 gene by the presence of the unique restriction enzyme sequence. Identifying the unique sequence of the inserted gene tag in combination with a vector sequence gave a fragment easily noticeable on the gel as presented in Figure 5 left top and bottom. The results indicated the presence and position of a particular sequence and its length from another that was confirmed by Sanger sequencing with home designed forward and reverse primers to cover twice the inserted genes. The list of restriction enzymes presents on the tag sequence and the pQβ7 vector was as follows: Pst I, Nhe I, and EcoR V. The successfully sequenced recombinant plasmids were obtained after confirming all positive restriction analyses and were used for phage expression and production.

#### 3.3.5. Recombinant Phage Morphology and Titer

As indicated elsewhere [8], the production of recombinant phage was conducted using an F^-^
*E. coli* HB101 avoiding premature evolutionary evens [8]. On the lawn of the indicator bacteria (K12 or Q13) after 12 h, the plaque sizes ranged from 0.4 to 2 mm in diameter in both wt and recombinant phages with similar titers (Figure 6; Table 2). The phages containing the truncated A1 were predominantly made up of small-size plaques while the rest with natural recombinant A1 were large in diameter. These results corroborated our early report and showed that all recombinant plasmids produced plaque-like phages on the lawn of indicator bacteria. The recombinant and wt phage titers were similar between 10^7^ and 10^9^ pfu/mL (Table 2) with exception of the phages exposing the His-tag that were improved after codon optimization and secondary RNA structure revisited (from 10^4^ to 10^7^ pfu/mL). This first-generation phage titer in F- bacteria was sufficient as a biosensor but for experimental purposes was further amplified and stored. Sanger sequencing, dot blotting, ELISA, and cryo-EM were used to confirm the genotype and phenotype of each recombinant phage.

#### 3.3.6. Genotype Analysis of Recombinant Phages

After six rounds of panning the phages extracted from a single plaque, the RNA was converted to cDNA, copied by RT-PCR, and sequenced. Initially, the secondary structure of native RNA compromised the RT-PCR, and intensive heat was applied to the RNA solution before the addition of the primer and reagents for reverse transcription (RT). Two sets of primer pairs were used to copy the cDNA region of the genome. The amplified cDNA was analyzed by agarose gel electrophoresis. The result is shown in Figure 5 top right with the appropriate band size of 250 bp (around the insert within the cloning cassette) and at bottom are the band size of 1.5 kb. The large fragment was sequenced for each recombinant phage and found to contain the appropriate sequence fused in frame with the A1 gene of the recombinant phage. Sequencing results confirmed the fragment sizes on an agarose gel and further confirmed that recombinant phage genome was obtained.

#### 3.3.7. Phenotype Analysis of Recombinant Phage

To analyze the probe (nanotag) presented on the recombinant phages, each stock of phages with three different dilutions was dialyzed and directly applied to a nitrocellulose membrane and probed with the appropriate antibody conjugated with HRP. The results showed a lineal presence of the nanotag on the spotted phages. The His-tag, the Strep II tag, and the newly developed biotin-tag reacted in a very similar way proportionately with the anti-His, streptavidin, and biotin all conjugated with HRP, respectively (Figure 7). The more dark spots on the membrane resulted in an increased presence, concentration, and titer of the canonical peptides (new phenotype) on the recombinant phage surface recognizing and binding to the appropriate target proteins compared to the wild type Qβ. The antibody and proteins conjugated to HRP revealed the new peptide displayed on the phage (new phenotype), which was confirmed by cryo-EM and ELISA. The dot blotting of FMDV epitopes with SD6 monoclonal antibody was performed elsewhere [8]. For high visibility, recombinant phages bearing the biotin-tag and Strep II tag were separately fused at the C-terminus with the His-tag and treated with 10 nm Ni-NTA-Nanogold which was then examined by electron microscope (Figure 8). The results showed the binding of the Nanogold to a His-tagged peptide on the phage (visible new phage, less than 50 nm diameter). The peptide tags were extended and detected with Ni-NTA-Nanogold confirming the desired phenotype of the recombinant phages.

#### 3.3.8. Competitive ELISA of Biotin-HRP and SD6 for Binding to QβBiotFMDV Recombinant Phage

Standard ELISA was initially carried with each tag or epitope to determine the saturated concentration of the antibody or appropriate recombinant phage proteins. Similar to our previous results [9,11], the recombinant phage titer of 10^7^ pfu/mL was coated and saturated the ELISA plates. Proteins were bound to the peptide tag as shown in Figure 9, with a hyperbolic curve, following a simple pattern of linear increase of binding to saturation. Next, an ELISA was performed with an increase concentration of SD6 vs. a saturated concentration of Biotin-HRP. The OD of the HRP product was recorded, plotted, and showed a decrease in HRP reactivity with increasing SD6 antibodies. Finally, our results showed that a potential binding of SD6 (anti-FMDV epitopes) prevented the biotin-HRP protein from accessing its appropriate transducer tag in a competitive manner. The tags and epitopes were accessible on the surface of the phage and were close to each other in the case of the fusion to promote detection and competition.

## 4. Discussion

Most phage display libraries and biosensor development technologies were exclusively designed and executed with the filamentous DNA phage M13 [38,39]. The M13 minor coat proteins (pIII) are located only at one end of the filamentous phage and are not equally distributed upon its surface similar to in the newly developed icosahedral Qβ RNA phage [8]. Additionally, due to this structural restriction, the M13 replication system is less adapted to evolutionary modifications relative to Qβ [8,9,10,11]. Recently, we successfully inserted and fused a 5-mer peptide library into the A1 minor coat protein of recombinant Qβ stably displayed. The FMDV epitope was selected, enriched, and amplified from the 5-mer library revealing a non-canonical epitope [8]. In this study, we extended the library size to 15 mer within the truncated A1 for broad selection of peptide and specific nanotag development. A titer of 10^9^ pfu/mL of recombinant phage was obtained with the 15-mer tag library inserted into the A1 and used to select HGHGWQIPVWPWGQG as a biotin-specific binding peptide. After many rounds of enrichment, the biotin recognizing peptide was compared to weak binder candidates to decipher an IPVW common motif. The selected tetrapeptide motif is the result of six rounds of biopanning under selective pressure by biotin binding to a spectrum of recombinant phage variants. Additionally, the enrichment of optimal variants was achieved by the application of elution by infection strategy, originally optimized in our group thereby avoiding the acidic treatment of selected phages that could affect and reduce their viability. By mimicking the natural infection of the phage to enable acid independent elution of the selected variant, a novel optimized toolkit of nanotechnology is validated for recombinant phage generation.

The selected biotin-binding peptide was subjected to comparative analysis with well-known peptides such as Strep II tag and His-tag together with peptides binding streptavidin and anti-His tag proteins, respectively, in order to learn more about its affinity to biotin. Our data showed that the tags presented separately on Qβ phage surface had similar affinity to their corresponding proteins. A maximum titer of 10^7^ pfu/mL of recombinant phages saturates the plate bottoms by binding to 2 µg of protein signal. A study had previously shown that a 15-amino-acid-long Strep II tag binds to streptavidin with a dissociation constant of 4 nM which could also be assigned to our novel biotin-binding peptide [14,40]. Similarly, a His-tag exposed on the surface of the recombinant phage showed high affinity for anti-His antibodies. Using these parallel studies, we term our novel biotin-binding peptide as nanotag since it is similar to the Strep II tag which has a small range of nM in its affinity for streptavidin. The dot blot analysis of the recombinant Qβ phages demonstrates the accessibility of the protein tag to the nano-tag probe exposed on the platform and confirms its potential with the level of affinity described. These findings remained the same when the nano-tag was inserted into a truncated or natural A1. The accessibility of a truncated A1 for this novel nanotechnology implies that less than half of this minor coat protein is anchored within the capsid shell of the recombinant phages with the probe position previously described on the T = 3 icosahedral shell of the Qβ phage. The truncated minor coat protein A1 therefore provides a room for larger peptide insertions to be displayed on the 12 corners of the recombinant geometric icosahedral phage, anchored less than halfway on the capsid shell, which is very accessible and assessable on the platform.

We have tested the fusion and exposition of pathogen-derived heteromeric peptides on the platform separated with a linker. However, in the case of the FMDV epitope and the His-tag, both were accessible only when presented at the extremity of other peptides or A1 (or C-terminus). On the other hand, the biotin-binding peptide (Biot-tag) and the Strep II tag affinities did not substantially change when presented at both positions of other peptides (N- or C-terminus). In contrast, both the FMDV epitope and His-tag affinities to their cognate antibodies were abolished when fused to the N-terminus of any of the nanotags, including the Au1 tag peptide [41]. This drastic change of the analyte SD6 in affinity to the FMDV epitope due to its position is probably a consequence of structural and conformational changes, which prevent accessibility and recognition by the cognate antibody and the abrogation of signals. These data highlight the importance of the structural conformation and the position of components in the recombinant Qβ phage as biosensors [42,43]. In addition to the RNA secondary structure during biosensor development, the probe and transducer peptide positions and structures should also be optimized. When a peptide used as a nanotag shows the same affinity in two different positions, this renders it a suitable candidate as a transducer. Therefore, the Strep II tag and Biot-tag were selected as transducer peptides in a quest for potential probes enabling the targeting of an FMDV derived epitope to its cognate monoclonal antibody called SD6. Additionally, peptide binding materials were investigated as transducers, notably peptide binding gold (Au1 and Au2), ZnS, and Co and were found to be useful in this newly developed nanotechnology. All peptide-binding materials were successfully displayed on the recombinant phage platform without affecting its viability. Moreover, the recombinant phages incubated with those nanoparticles were not affected and could still recognize, adsorb, and infect the *E. coli* K12 and Q13 hosts. This result confirms the resistance and usefulness of this phage to harsh environmental conditions without the need for special equipment for maintenance. Interestingly, recombinant Qβ phage has the potential to display peptides that bind inorganic materials, thereby broadening the range of applications of this RNA phage-based biosensor display platform.

The current RNA phage library can be used to target a mixture of known and unknown entities enabling the selected genotype of the probe(s) to be eluted through infection and characterized unlike the M13 phage display strategy which is usually limited by steps involving acidic treatment. Once the probe peptide fusion and positioning with the transducer peptide are achieved, newly discovered and unknown entities can be concentrated prior to their identification. There seem to be little or no limitations to this novel nanotechnology when applied to the detection of bio-threat agents as they often occur in harsh environments and conditions. Thus, our nanotechnology holds a high potential for developing RNA-based phage biosensor platforms for Homeland Security and the Department of Defense. The truncated A1 on a viable phage within the cloning cassette region gives more room for large sequence insertion, compared to the wild type, and could be used to display small camel or shark antibodies, particularly VHH nanobodies [44]. A repertoire of naïve nanobodies displayed on Qβ phage can serve as a library of constructed nanobodies and be used to probe an array of important biomedical elements for their cognate agonist [45]. In contrast to the poor stability that conventional antibodies may suffer, biosensors with nanobodies will be strengthened by the tremendous resistance of recombinant RNA phage display platforms. ELISA was used to determine the quantitative parameters of each of the nanotags either separately or in combination with the recombinant phage platform together with the positioning of probe or transducer peptides. In addition, since the saturation concentrations of the nanotags are well known, the potential for designing competitive analyses or assays becomes relevant for our nanotechnology platform. In this light, an FMDV-specific immunogenic epitope in a competitive titration assay with its cognate SD6 antibody was used to achieve binding and quantitative recognition of the target. The detection of FMDV specific antibodies with a recombinant RNA phage platform ushers in a new era in biosensor nanotechnology where the transducer could change based on the targeted agonist and its environmental conditions. Moreover, dependent upon the proximity of the probe and transducer peptides, binding to one cognate antibody abrogated access by its competitor, thereby providing a novel platform for combined qualitative and quantitative analysis of nanobioprobes [46,47]. Thus, the standardized transducer (biotin-binding peptide) can be used to titer unknown cognate antibody probes using a similar competitive ELISA as demonstrated in our study. Organic transducers can be used for sensitive biomolecular targets and inorganic substance application under harsh conditions. Regarding existing and continuously evolving pathogens like SARS-CoV-2, Lassa fever virus, Ebola, Marburg, and monkeypox-derived epitopes, we envisaged developing biosensors using a Strep II tag or biotin-binding peptides for the titration of pathogen-specific protective antibodies in vaccinated and exposed populations over time. Similar biosensors could be developed for the clinical detection of emergent and re-emergent pathogens.

## 5. Conclusions

To the best of our knowledge, this is the first example of the use of an evolutionary RNA phage display library and peptides for biosensor development. The results obtained here may pave the way and open new strategies for developing sensing probes for biomedical, security, and environmental sample applications. This novel technology can be used in the future to revolutionize the application of RNA phage display for nanobodies and single chain antibodies in biosensor development. Work needs to be conducted on the correlation between this phage platform titer, the total recombinant RNA concentration, the probe quantity exposed on A1, and the minimum and maximum binding capacities in solution compared to an immobilized surface to establish all the parameters of this toolkit. Other sensing schemes based on inorganic transduction techniques should be introduced within this platform.

## Figures and Tables

**Figure 1 viruses-15-01414-f001:**
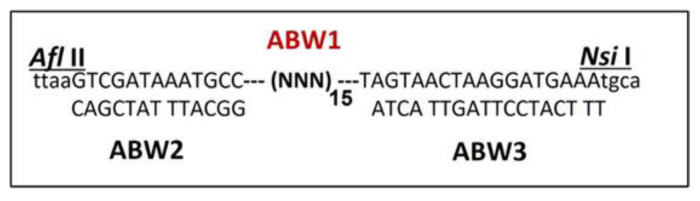
Design and schematic representation of the nano-tag oligonucleotide library sequences. ABW1 is the randomized sequence synthesis for library generation and population of variant phages production against proteins and materials selection. ABW2 sequence complementary to the N-terminus of the library ABW1 flanked Afl II restriction enzyme sequence. ABW3 sequence complementary to the C-terminus of the library ABW1 flanked Nsi I restriction enzyme sequence.

**Figure 2 viruses-15-01414-f002:**
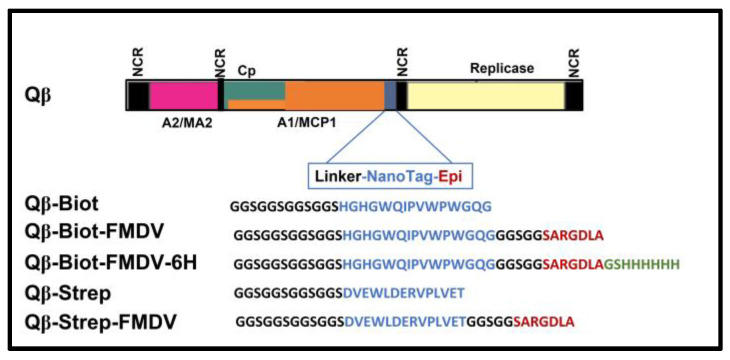
Schematic representation of the general organization of the recombinant Qβ with tags (amino acids in the blue color). From up to down QβA1Biot phages with the biotin tag; QβA1BiotFMDV phages with FMDV epitope separated by biotin; QβA1BiotFMDV6H is the same as previous ending with 6xhis-tag; QβA1Strep phages with strep II tag; QβA1StrepFMDV are same as previous but with the FMDV epitope. The amino acids in black represent the linker; in red is the FMDV epitope; and in green is the 6xHis-tag, respectively. NCR is the non-coding region; Cp is the coat protein gene region.

**Figure 3 viruses-15-01414-f003:**
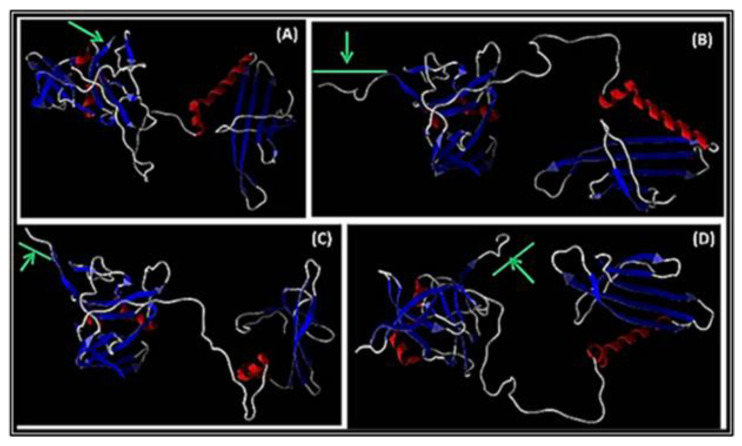
Three-dimensional representation of structures prediction of the wild type and recombinant Qβ minor coat protein (MCP) A1. In (**A**) Qβ wild type A1; (**B**) QβA1Biot; (**C**) QβA1Strep; (**D**) QβA1Au. The green arrow at the C-terminus region of the A1 is an indication of the A1 insertion area or the inserted peptides. The red and blue colored parts of the A1 structures indicate α-helix and β-sheets of the N terminus, respectively.

**Figure 4 viruses-15-01414-f004:**
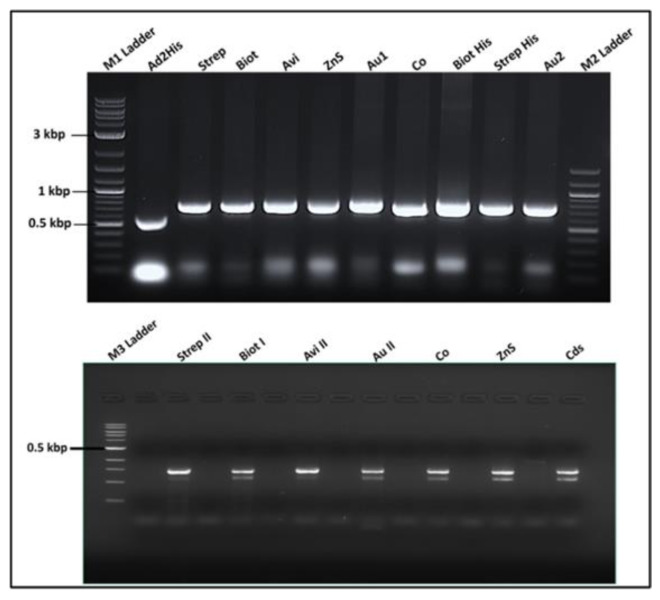
Unpurified PCR products of the nanotag sequences fused with A1 minor coat protein gene. Top gel represents the fragment of the A1 (600 bp starting Bpu 10I) fused with the nanotag gene such as A1 deleted (−150 bp) with 6xhis-tag; Strep II tag (Strep); biotin tag (Biot); avidin tag (Avi); ZnS tag (ZnS); gold tag (Au1, Au2). Bottom gel is the same as previous with A1 (250 bp starting Afl II) fused with nanotags.

**Figure 5 viruses-15-01414-f005:**
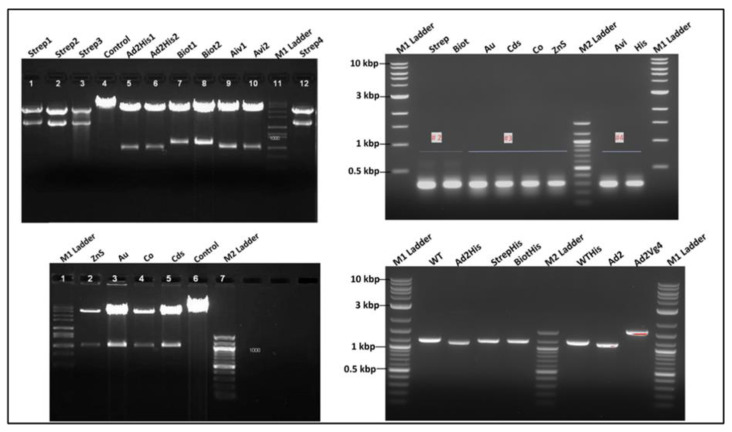
Images of the agarose gel electrophoresis of the Qβ phage display system vector construction analysis. On the top left: product of the recombinant pQβ7 restriction digestion lane 1–3 and 12 pQβStrep with Pst I; lane 4 pQβWT digested Pst I; lane 5–6 pQβAd2 digested Nhe I; lane 7–8 pQβBiot digested Nhe I; lane 9–10 pQβAvi digested EcoR V. On the bottom left: lane 2–5 pQβZns, pQβAu, pQβCo, pQβCds digested Nhe I, respectively; lane 6 pQβ7. On the right is the RT-PCR of RNA extracted from purified phages Qβstrep, QβBiot, QβAu, QβCds, QβCo, QβZnS, QβAvi, QβHis, Qβ7, QβAd2His, QβAd2, QβAd2VG4, respectively, on the top being the fragment amplified between Afl II and Nsi I and at the bottom between Bpu 10I and Nsi I. M1 and M2 ladders are 100 bp and 10 kbp, respectively.

**Figure 6 viruses-15-01414-f006:**
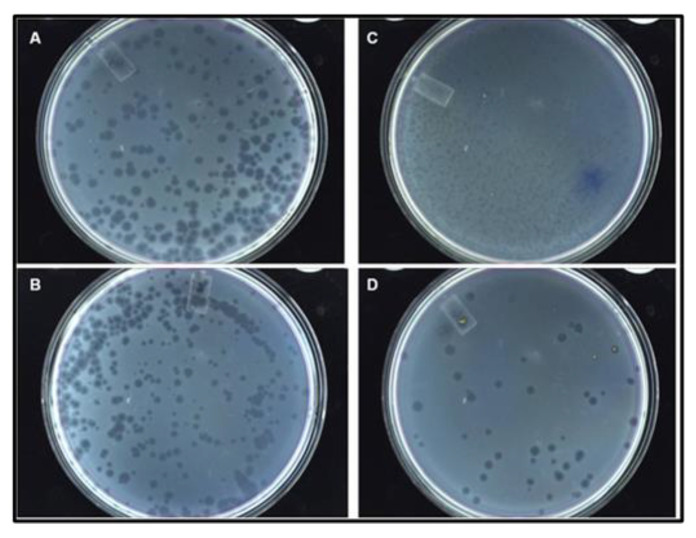
Morphology of the wild type vs. recombinant Qβ phage plaques on the Q13 *E. coli* lawn. (**A**) Qβ; (**B**) QβAd2; (**C**) QβBiot; (**D**) QβStrep. The experiment was conducted at multiplicity of infection of 2, and plates were 12 h old.

**Figure 7 viruses-15-01414-f007:**
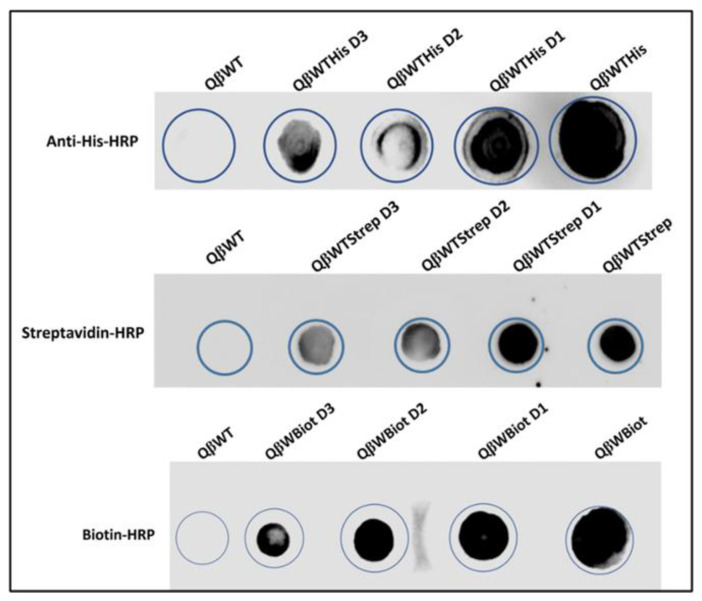
Dot blotting of the recombinant purified QβBiot, QβStrep, and QβHis phages and QβWT (Qβ wild type as a control). D1 to D3 are the 10^10^, 10^9^, and 10^8^ pfu/mL of phages spotted and probed with anti-His-HRP, anti-Strep-HRP, and anti-Biot-HRP directly and, respectively. All phages were dialyzed against phage buffer.

**Figure 8 viruses-15-01414-f008:**
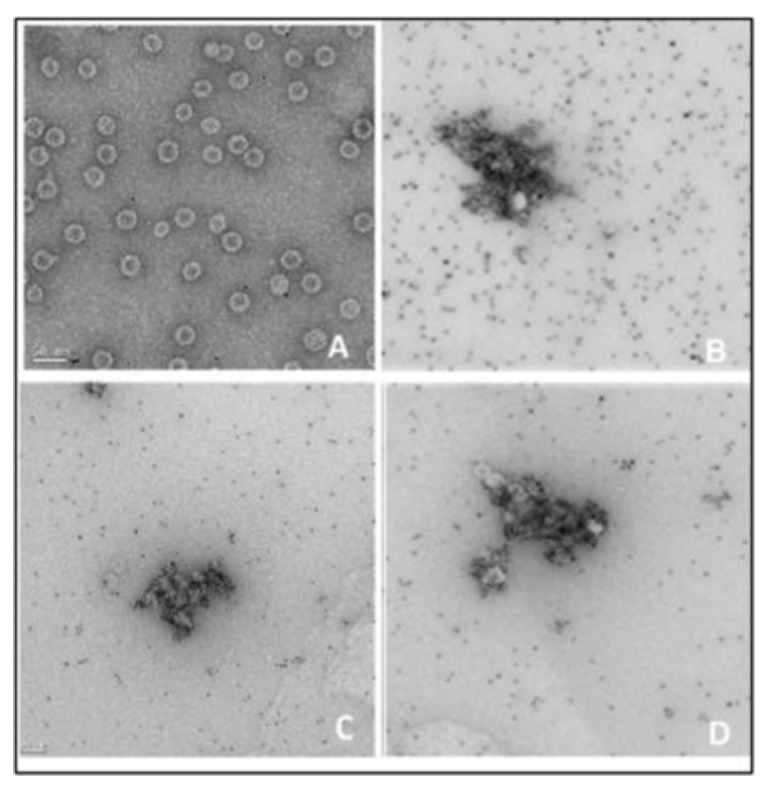
Cryo-electron microscope image of recombinant phages detected with 10 nm nickeled gold (Ni-NTA-Nanogold). (**A**) Qβ7 wild type control; (**B**) QβStrepHis; (**C**) QβHis; (**D**) QβBiotHis.

**Figure 9 viruses-15-01414-f009:**
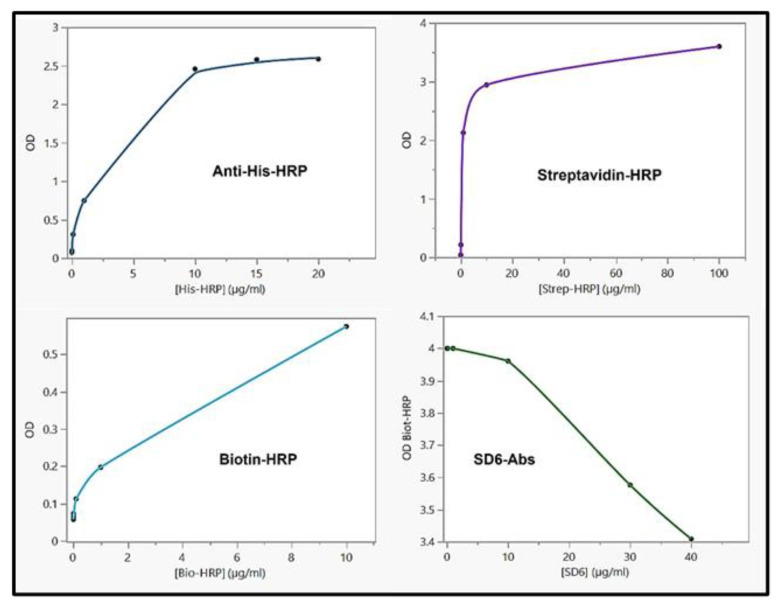
ELISA with recombinant phages. Anti-His-HRP: the plot analysis of optical density (OD) of the horseradish peroxidase (HRP) product vs. increase of concentration of anti-His-HRP antibody (plate coated with QβHis); Streptavidin-HRP: the plot analysis of OD vs. increase of concentration of streptavidin conjugated to HRP (plate coated with QβStrep); Biotin-HRP: the plot analysis of OD vs. increase of concentration of biotin conjugated to HRP (plate coated with QβBiot); SD6-Abs: plot analysis of OD vs. increase of concentration of SD6 antibody against FMDV immunogenic epitope (plate coated with QβBiotFMDV) with a fix concentration of Biotin-HRP (10 µg/mL).

**Table 1 viruses-15-01414-t001:** List of oligonucleotides names and sequences with bold are the phages A1 gene portions.

Names	Primer–DNA Sequences
**ABW1**	**ttaaGTCGATAAATGCC** (NNN)15 **TAGTAACTAAGGATGAAAtgca**
**ABW2**	**CAGCTATTACGG**
**ABW3**	**ATCATTGATTCCTACTTT**
**Au1**	**AATGTCCAATTCAAGCTGTGATAGTCGTTCCTCGTGCT**gaattCgtcagtggttcctctcccgacagttag**TAActaaggatgaaatgcATGgg**
**Au2**	**GTCCAATTCAAGCTGTGATAGTCGTTCCTCGTaaGCTtacaggtacttcagtcctcattgcaactccatacgtttagTAActaaggatgaaatgcATGgg**
**Silca**	**GTCCAATTCAAGCTGTGATAGTCGTTCCTCGTaaGCTt**atgagccctcaccctcatccgcgacaccatcacacc**TAGTAActaaggatgaaatgcATGgg**
**Cds**	**aatgtccaattcaagctgtgatagtcgttcctcgtgct**AGCCTGACCCCGCTGACCACCAGCCATCTGCGCAGC**tagTAActaaggatgaaatgcATGgg**
**ZnS**	**tgtccaattcaagctgtgatagtcgttcctcgtgct**GTGATTAGCAACCATGCGGGCAGCAGCCGCCGCCTG**tagTAActaagCTTgatgaaatgcATGT**
**6xhis-tag**	**gtccaattcaagctgtgatagtcgttcctcgtgctGGTCATCACCATCATCATCACGGGTCC**tagtaaGCTAGCctaaggatgaaatgcatgtgg
**Biotin-tag**	**Gtccaattcaagctgtgatagtcgttcctcgtgcagcggcca**tcatcatcatcatcatggcagc**TAGTAAGCTAGCctaaggatgaaatgcatgtgg**
**Strep II-tag**	**Gtccaattcaagctgtgatagtcgttcctcgtgc**Gatgtggaatggctggatgaacgcgtgccgctggtggaaacc**TAGTAActaagCTTgatgaaatgcATGT**
**Co**	**aaatgtccaattcaagctgtgatagtcgttcctcgtgct**GCTagcGAAGAAGAAGAAtagTAA**ctaaggatgaaatgcATGTCTAA**

**Table 2 viruses-15-01414-t002:** Comparison of recombinant phage titers from the first generation (with *E. coli* HB101) to the rounds of host infection (2nd and 3rd).

Host/Phages	Qβ	QβHis	QβStrep	QβBiot
*E. coli* HB101(1st)	10^9^ pfu/mL	10^8^ pfu/mL	10^7^ pfu/mL	10^7^ pfu/mL
*E. coli* Q13 (2nd)	10^12^ pfu/mL	10^10^ pfu/mL	10^9^ pfu/mL	10^9^ pfu/mL
*E. coli* K12 (3rd)	10^14^ pfu/mL	10^12^ pfu/mL	10^11^ pfu/mL	10^11^ pfu/mL

## Data Availability

Data supporting the reported results can be found at Addgene (https://www.addgene.org, accessed on 9 May 2023) and for depositing plasmids at Addgene our Deposit Number is: 82735.

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
