# Peer review of "Evolutionary Qβ Phage Displayed Nanotag Library and Peptides for Biosensing"

_viruses, 2023, doi:10.3390/v15071414_

Round 1

Reviewer 1 Report

This work presents a novel approach in biosensor development using an evolutionary RNA Qβ peptide phage display library and heterodimeric peptides. The authors claim that this is the first example of such an application. The technology described is innovative and has the potential to revolutionize the use of RNA phage display for nanobodies and single chain antibodies in biosensor development. Additionally, the authors suggest incorporating other sensing schemes based on inorganic transduction techniques into this platform.

Overall, the results obtained from this research is significant and could potentially lead to the development of sensing probes for biomedical, security, and environmental sample applications, offering new strategies in biosensor development.

The manuscript is well written and the data presented clearly.

There were a few places in the manuscript with minor English errors, for example, on page8, line 368: "...used to bound" should be "...used to bind"; on the same page, line 371: "...weaker binder" should be changed to "...weaker binders", etc. 

Author Response

All concerns were addressed in the manuscript attached here with a cover letter.

Reviewer 2 Report

Discussion long and unstructured, speculations about the possible application of described technology, not supported by experiments. 

Check the proper use of pfu/ml and titration instead of using the appropriate amount of phage pfu. 

Give an explanation for terms like probe and transducer, and consider the shortening of sentences. 

Consider the change of title of the manuscript - it is not very focused, it is diffused (what about the "nanotag library"...the application of novel technology to select specific nanotag is presented;  "peptide binding materials" are not described, just noted).  

Explain please what was the "starting sequence" inserted in truncated A1. 

The Wt phage was not used for engineering, however, such a notion is present. 

Avoid the term RNA-display instead of Qbeta phage display.   

Problems with English grammar, a lot of mistakes, wrong word order, etc. Use the help of appropriate services for the editing of the manuscript (Grammarly,  for example).  

Author Response

All the comments are addressed, and the corrected manuscript is attached here.

Sincerely
